# Nonstationary Sparse Spectral Permanental Process

**Zicheng Sun**[1][§], **Yixuan Zhang**[2][§], **Zenan Ling**[3], **Xuhui Fan**[4] **Feng Zhou**[1,5][*]

[1]Center for Applied Statistics and School of Statistics, Renmin University of China
[2]School of Statistics and Data Science, Southeast University
[3]School of EIC, Huazhong University of Science and Technology
[4]School of Computing, Macquarie University
[5]Beijing Advanced Innovation Center for Future Blockchain and Privacy Computing
`{sunzicheng2020, feng.zhou}@ruc.edu.cn, zh1xuan@hotmail.com`

## Abstract

Existing permanental processes often impose constraints on kernel types or stationarity, limiting the model's expressiveness. To overcome these limitations, we propose a novel approach utilizing the sparse spectral representation of nonstationary kernels. This technique relaxes the constraints on kernel types and stationarity, allowing for more flexible modeling while reducing computational complexity to the linear level. Additionally, we introduce a deep kernel variant by hierarchically stacking multiple spectral feature mappings, further enhancing the model's expressiveness to capture complex patterns in data. Experimental results on both synthetic and real-world datasets demonstrate the effectiveness of our approach, particularly in scenarios with pronounced data nonstationarity. Additionally, ablation studies are conducted to provide insights into the impact of various hyperparameters on model performance. Code is publicly available at https://github.com/SZC20/DNSSPP.

## 1   Introduction

Many application domains involve point process data, which records the times or locations of events occurring within a region. Point process models are used to analyze these event data, aiming to uncover patterns of event occurrences. The Poisson process [15], as an important model in the field of point processes, plays a significant role in neuroscience [4], finance [11], criminology [31], epidemiology [5], and seismology [8]. Traditional Poisson processes assume parameterized intensity functions, which severely restricts the flexibility of the model. To address this issue, a viable solution is to employ Bayesian nonparametric methods, by imposing a nonparametric Gaussian process (GP) prior on the intensity function, resulting in the Gaussian Cox process [3]. This greatly enhances the flexibility of the model, while also endowing it with uncertainty quantification capabilities.

In Gaussian Cox processes, posterior inference of the intensity is challenging because the GP prior is not conjugate to the Poisson process likelihood that includes an intensity integral. Furthermore, to ensure the non-negativity of the intensity, we need to use a link function to transform the GP prior. Commonly used link functions include exponential [20; 12], sigmoid [1; 9; 6], square [18; 7; 29], ReLU [16], and softplus [24]. Among these, using the square link function, i.e., the permanental process [19], allows for the analytical computation of the intensity integral [18; 33; 29], thus receiving widespread attention in recent years. For this reason, this work focuses on the permanental process.

Currently, the permanental process faces three issues: **(1)** The permanental process inherits the notorious cubic computational complexity of GPs, making it impractical for use with a large amount

---

[*]Corresponding author.
[§]Equal contributions.

38th Conference on Neural Information Processing Systems (NeurIPS 2024).

of data [18; 14]. **(2)** Existing works on permanental processes either require certain standard types of kernels, such as the squared exponential kernel, etc., to ensure that the intensity integral has an analytical solution [18; 7; 33]; or they require the kernels to be stationary [14; 29]. These constraints limit the expressive power of the model. **(3)** Furthermore, existing works have predominantly utilized simple shallow kernels, which restricts the flexibility of the model. Although some shallow kernels [34; 28] are flexible and theoretically capable of approximating any bounded kernel, they are limited in representing complex kernels due to computational constraints in practical usage.

In this study, we utilize the sparse spectral representation of nonstationary kernels to address these limitations in modeling the permanental process. The sparse spectral representation provides a low-rank approximation of the kernel, effectively reducing the computational complexity from cubic to linear level. The nonstationary sparse spectral kernel overcomes the limitation of stationary assumption. By treating the frequencies as kernel parameters for optimization, we can directly learn the nonstationary kernel from the data in a flexible manner without restricting the kernel's form. We further extend the shallow nonstationary sparse spectral kernel by hierarchically stacking multiple spectral feature mappings to construct a deep kernel, which exhibits significantly enhanced expressive power compared to shallow ones. We term the constructed model as Nonstationary Sparse Spectral Permanental Process (NSSPP) and the corresponding deep kernel variant as DNSSPP.

We conduct experiments on both synthetic and real-world datasets. The results indicate that when the data is (approximately) stationary, (D)NSSPP achieves similar performance to stationary baselines. However, when the nonstationarity in the data is pronounced, (D)NSSPP can outperform baseline models, demonstrating the superiority of (D)NSSPP. Additionally, we perform ablation studies to assess the impact of various model hyperparameters.

## 2 Related Work

In this section, we present some related works on how to reduce the computational complexity of the stationary permanental process, as well as some research on nonstationary kernels.

**Efficient Permanental Process**   Due to the adoption of the GP prior in the permanental process, posterior inference involves computing the inverse of the kernel matrix. This results in a computational complexity of $\mathcal{O}(N^3)$, where $N$ is the number of data points. To reduce computational complexity, several works have introduced low-rank approximation methods from the GP domain into the permanental process, such as inducing points [18], Nyström approximation [7; 33], and spectral representation [14; 29]. These methods essentially involve low-rank approximations of the kernel matrix, thereby reducing the computational complexity from $\mathcal{O}(N^3)$ to $\mathcal{O}(NR^2)$, where $R \ll N$ is the number of rank, such as the number of inducing points or frequencies. In this work, we utilize the spectral representation method, reducing the computational complexity from cubic to linear level. Another reason for adopting the spectral representation is that it facilitates the construction of the subsequent deep nonstationary kernel.

**Nonstationary Kernel**   Stationary kernels assume that the similarity between different locations depends only on their relative distance. However, previous studies have indicated that the stationary assumption is not suitable for dynamic complex tasks, as the similarity is not consistent across the input space [23]. To address this issue, some works proposed nonstationary kernels by transforming the input space [30] or using input-dependent parameters [10]. Additionally, other works leveraged the generalized Fourier transform of kernels [37] to propose nonstationary spectral kernels [26; 32]. Although the aforementioned shallow nonstationary kernels are flexible and theoretically capable of approximating any bounded kernels, they are limited in representing complex kernels in practical usage. In recent years, many deep architectures have been introduced into kernels, significantly enhancing their expressive power [35; 36]. Despite the development of (deep) nonstationary kernels in GP, interestingly, to the best of our knowledge, there has been no prior work applying them in Gaussian Cox processes. This gap is what this work seeks to address.

## 3 Preliminaries

In this section, we provide some fundamental knowledge about the permanental process and sparse spectral kernel.

## 3.1 Permanental Process

For a Poisson process, if $\mathbf{x} \in \mathcal{X} \subset \mathbb{R}^D$, the intensity function $\lambda(\mathbf{x}) = \lim_{\delta_\mathbf{x} \to 0} \mathbb{E}[N([\mathbf{x}, \mathbf{x} + \delta_\mathbf{x}])]/\delta_\mathbf{x}$ can be used to quantify the rate of events occurring at location $\mathbf{x}$. A Cox process can be viewed as a Poisson process whose intensity function itself is a random process. The Gaussian Cox process employs the GP to model the intensity function: $\lambda(\mathbf{x}) = l \circ f(\mathbf{x})$ where $f$ is a GP function and $l : \mathbb{R} \to \mathbb{R}^+$ is a link function to ensure the non-negativity of the intensity function.

The nonparametric nature of GP enables the Gaussian Cox process to be flexible in modeling events in space or time. However, the posterior inference of Gaussian Cox process is challenging. According to the likelihood function of a Poisson process:

$$p(\{\mathbf{x}_i\}_{i=1}^N \,|\, \lambda(\mathbf{x}) = l \circ f(\mathbf{x})) = \prod_{i=1}^N \lambda(\mathbf{x}_i) \exp\left(- \int_\mathcal{X} \lambda(\mathbf{x}) d\mathbf{x}\right), \tag{1}$$

and the Bayesian framework, the posterior of latent function $f$ can be written as:

$$p(f \,|\, \{\mathbf{x}_i\}_{i=1}^N) = \frac{\prod_{i=1}^N \lambda(\mathbf{x}_i) \exp(- \int_\mathcal{X} \lambda(\mathbf{x}) d\mathbf{x}) \mathcal{GP}(f|0, k)}{\int \prod_{i=1}^N \lambda(\mathbf{x}_i) \exp(- \int_\mathcal{X} \lambda(\mathbf{x}) d\mathbf{x}) \mathcal{GP}(f|0, k) df}. \tag{2}$$

In Eq. (2), there are two integrals. The first one $\int_\mathcal{X} \lambda(\mathbf{x}) d\mathbf{x}$ cannot be solved due to the stochastic nature of $f$. Additionally, the integral over $f$ in the denominator is a functional integral in the function space, which is also infeasible. This issue is the well-known doubly-intractable problem.

The link function $l : \mathbb{R} \to \mathbb{R}^+$ has many choices, such as exponential, sigmoid, square, ReLU, and softplus. This work focuses on the utilization of the square link function, also known as the permanental process. Furthermore, if we set the kernel in the GP prior to be stationary: $k(\mathbf{x}_1, \mathbf{x}_2) = k(\mathbf{x}_1 - \mathbf{x}_2)$, then the model is referred to as a stationary permanental process.

## 3.2 Sparse Spectral Kernel

The sparse spectral representation is a common method for the low-rank approximation of kernels. This approach is based on Bochner's theorem.

**Theorem 3.1.** *[2] A stationary kernel function $k(\mathbf{x}_1, \mathbf{x}_2) = k(\mathbf{x}_1 - \mathbf{x}_2) : \mathbb{R}^D \to \mathbb{R}$ is bounded, continuous, and positive definite if and only if it can be represented as:*

$$k(\mathbf{x}_1 - \mathbf{x}_2) = \int_{\mathbb{R}^D} \exp(i\boldsymbol{\omega}^\top(\mathbf{x}_1 - \mathbf{x}_2)) d\mu(\boldsymbol{\omega}),$$

*where $\mu(\boldsymbol{\omega})$ is a bounded non-negative measure associated to the spectral density $p(\boldsymbol{\omega}) = \frac{\mu(\boldsymbol{\omega})}{\mu(\mathbb{R}^D)}$.*

Defining $\phi(\mathbf{x}) = \exp(i\boldsymbol{\omega}^\top \mathbf{x})$, we have $k(\mathbf{x}_1 - \mathbf{x}_2) = \sigma^2 \mathbb{E}_{p(\boldsymbol{\omega})}[\phi(\mathbf{x}_1)\phi(\mathbf{x}_2)^*]$ where $*$ denotes the complex conjugate and $\sigma^2 = \mu(\mathbb{R}^D)$. If we utilize the Monte Carlo to sample $\{\boldsymbol{\omega}_r\}_{r=1}^R$ independently from $p(\boldsymbol{\omega})$, then the kernel can be approximated by

$$k(\mathbf{x}_1 - \mathbf{x}_2) \approx \frac{\sigma^2}{R} \sum_{r=1}^R \phi_r(\mathbf{x}_1)\phi_r(\mathbf{x}_2)^* = \Phi^{(R)}(\mathbf{x}_1)^\top \Phi^{(R)}(\mathbf{x}_2)^*, \tag{3}$$

where $\phi_r(\mathbf{x}) = \exp(i\boldsymbol{\omega}_r^\top \mathbf{x})$, $\Phi^{(R)}(\mathbf{x}) = \frac{\sigma}{\sqrt{R}}[\phi_1(\mathbf{x}), \cdots, \phi_R(\mathbf{x})]^\top$, which corresponds to the random Fourier features (RFF) proposed by [25]. It can be proved that Eq. (3) is equivalent to a $2R$-sized trigonometric representation (proof is provided in Appendix A):

$$\Phi^{(R)}(\mathbf{x}) = \frac{\sigma}{\sqrt{R}}[\cos(\boldsymbol{\omega}_1^\top \mathbf{x}), \cdots, \cos(\boldsymbol{\omega}_R^\top \mathbf{x}), \sin(\boldsymbol{\omega}_1^\top \mathbf{x}), \cdots, \sin(\boldsymbol{\omega}_R^\top \mathbf{x})]^\top. \tag{4}$$

When we specify the functional form of the kernel (spectral measure $\mu$), the RFF method can learn the parameters of the kernel (spectral measure $\mu$) from the data. However, when the functional form of the kernel (spectral measure $\mu$) is unknown, RFF can not flexibly learn the kernel (spectral measure $\mu$) from the data. Another approach is to treat the frequencies $\{\boldsymbol{\omega}_r\}_{r=1}^R$ as kernel parameters. By optimizing these frequencies, we can directly learn the kernel from the data in a flexible manner, which corresponds to the sparse spectrum kernel method proposed by [17].

# 4 Our Model

The permanental process inherits the cubic computational complexity of GPs, rendering it impractical for large-scale datasets. To efficiently conduct posterior inference, many works employed various low-rank approximation methods [18; 7; 33]. These methods require certain standard types of kernels, such as the squared exponential kernel, polynomial kernel, etc., to ensure that the intensity integral has analytical solutions. This limitation severely affects the flexibility of the model. To address this issue, sparse spectral permanental processes are proposed in recent years [14; 29]. The advantage of this method lies in its ability to analytically compute the intensity integral, while not requiring restrictions on the kernel's functional form. However, the drawback is that it is only applicable to stationary kernels. Therefore, a natural question arises: can we further generalize the sparse spectral permanental processes to not only remove restrictions on the kernel's functional form but also make them applicable to nonstationary kernels?

## 4.1 Nonstationary Sparse Spectral Permanental Process

Existing sparse spectral permanental processes rely on the prerequisite of Bochner's theorem, which limits the extension of the sparse spectral method to nonstationary kernels. Fortunately, for more general nonstationary kernels, the spectral representation similar to Bochner's theorem also exists.

**Theorem 4.1.** *[37] A nonstationary kernel function $k(\mathbf{x}_1, \mathbf{x}_2) : \mathbb{R}^D \times \mathbb{R}^D \to \mathbb{R}$ is bounded, continuous, and positive definite if and only if it can be represented as:*

$$k(\mathbf{x}_1, \mathbf{x}_2) = \int_{\mathbb{R}^D \times \mathbb{R}^D} \exp\left(i(\boldsymbol{\omega}_1^\top \mathbf{x}_1 - \boldsymbol{\omega}_2^\top \mathbf{x}_2)\right) d\mu(\boldsymbol{\omega}_1, \boldsymbol{\omega}_2),$$

*where $\mu(\boldsymbol{\omega}_1, \boldsymbol{\omega}_2)$ is a Lebesgue-Stieltjes measure associated to some bounded positive semi-definite spectral density $p(\boldsymbol{\omega}_1, \boldsymbol{\omega}_2) = \frac{\mu(\boldsymbol{\omega}_1, \boldsymbol{\omega}_2)}{\mu(\mathbb{R}^D, \mathbb{R}^D)}$.*

Similarly, we can use the Monte Carlo method to approximate it. However, it is worth noting that, to ensure that the spectral density $p(\boldsymbol{\omega}_1, \boldsymbol{\omega}_2)$ is positive semi-definite, we must first symmetrize it, i.e., $p(\boldsymbol{\omega}_1, \boldsymbol{\omega}_2) = p(\boldsymbol{\omega}_2, \boldsymbol{\omega}_1)$; and secondly introduce diagonal components $p(\boldsymbol{\omega}_1, \boldsymbol{\omega}_1)$ and $p(\boldsymbol{\omega}_2, \boldsymbol{\omega}_2)$. We can take a general density function $g$ on the product space and then parameterize $p(\boldsymbol{\omega}_1, \boldsymbol{\omega}_2)$ in the following form to ensure its positive semi-definite property [26; 32]:

$$p(\boldsymbol{\omega}_1, \boldsymbol{\omega}_2) = \frac{1}{4}(g(\boldsymbol{\omega}_1, \boldsymbol{\omega}_2) + g(\boldsymbol{\omega}_2, \boldsymbol{\omega}_1) + g(\boldsymbol{\omega}_1, \boldsymbol{\omega}_1) + g(\boldsymbol{\omega}_2, \boldsymbol{\omega}_2)). \tag{5}$$

Then, we can obtain the sparse spectral feature representation of any nonstationary kernel:

$$
\begin{aligned}
k(\mathbf{x}_1, \mathbf{x}_2) &= \frac{\sigma^2}{4} \mathbb{E}_{g(\boldsymbol{\omega}_1, \boldsymbol{\omega}_2)} \left[ \exp\left(i(\boldsymbol{\omega}_1^\top \mathbf{x}_1 - \boldsymbol{\omega}_2^\top \mathbf{x}_2)\right) + \exp\left(i(\boldsymbol{\omega}_2^\top \mathbf{x}_1 - \boldsymbol{\omega}_1^\top \mathbf{x}_2)\right) \right. \\
&\quad \left. + \exp\left(i(\boldsymbol{\omega}_1^\top \mathbf{x}_1 - \boldsymbol{\omega}_1^\top \mathbf{x}_2)\right) + \exp\left(i(\boldsymbol{\omega}_2^\top \mathbf{x}_1 - \boldsymbol{\omega}_2^\top \mathbf{x}_2)\right) \right] \\
&\approx \frac{\sigma^2}{4R} \sum_{r=1}^{R} \left[ \exp\left(i(\boldsymbol{\omega}_{1r}^\top \mathbf{x}_1 - \boldsymbol{\omega}_{2r}^\top \mathbf{x}_2)\right) + \ldots + \exp\left(i(\boldsymbol{\omega}_{2r}^\top \mathbf{x}_1 - \boldsymbol{\omega}_{2r}^\top \mathbf{x}_2)\right) \right] \\
&= \Phi_1^{(R)}(\mathbf{x}_1)^\top \Phi_2^{(R)}(\mathbf{x}_2) + \Phi_2^{(R)}(\mathbf{x}_1)^\top \Phi_1^{(R)}(\mathbf{x}_2) + \Phi_1^{(R)}(\mathbf{x}_1)^\top \Phi_1^{(R)}(\mathbf{x}_2) + \Phi_2^{(R)}(\mathbf{x}_1)^\top \Phi_2^{(R)}(\mathbf{x}_2) \\
&= \boldsymbol{\varphi}^{(R)}(\mathbf{x}_1)^\top \boldsymbol{\varphi}^{(R)}(\mathbf{x}_2),
\end{aligned}
\tag{6}
$$

where $\sigma^2 = \mu(\mathbb{R}^D, \mathbb{R}^D)$, $\{\boldsymbol{\omega}_{1r}, \boldsymbol{\omega}_{2r}\}_{r=1}^R$ are independent samples from $g(\boldsymbol{\omega}_1, \boldsymbol{\omega}_2)$, and

$$
\begin{aligned}
\Phi_1^{(R)}(\mathbf{x}) &= \frac{\sigma}{2\sqrt{R}}[\cos(\boldsymbol{\omega}_{11}^\top \mathbf{x}), \cdots, \cos(\boldsymbol{\omega}_{1R}^\top \mathbf{x}), \sin(\boldsymbol{\omega}_{11}^\top \mathbf{x}), \cdots, \sin(\boldsymbol{\omega}_{1R}^\top \mathbf{x})]^\top, \\
\Phi_2^{(R)}(\mathbf{x}) &= \frac{\sigma}{2\sqrt{R}}[\cos(\boldsymbol{\omega}_{21}^\top \mathbf{x}), \cdots, \cos(\boldsymbol{\omega}_{2R}^\top \mathbf{x}), \sin(\boldsymbol{\omega}_{21}^\top \mathbf{x}), \cdots, \sin(\boldsymbol{\omega}_{2R}^\top \mathbf{x})]^\top, \\
\boldsymbol{\varphi}^{(R)}(\mathbf{x}) &= \Phi_1^{(R)}(\mathbf{x}) + \Phi_2^{(R)}(\mathbf{x}).
\end{aligned}
\tag{7}
$$

If the GP's kernel in the permanental process (Eq. (2)) adopts the above sparse spectral nonstationary kernel, then we refer to this model as NSSPP.

## 4.2 Deep Nonstationary Sparse Spectral Permanental Process

We can further enhance the expressive power of the nonstationary kernel by stacking multiple spectral feature mappings hierarchically to construct a deep kernel. The spectral feature mapping $\varphi^{(R)}(\mathbf{x})$ has another equivalent form:

$$\varphi^{(R)}(\mathbf{x}) = \frac{\sigma}{\sqrt{2R}}[\cos(\boldsymbol{\omega}_{11}^\top \mathbf{x} + b_{11}) + \cos(\boldsymbol{\omega}_{21}^\top \mathbf{x} + b_{21}), \ldots, \cos(\boldsymbol{\omega}_{1R}^\top \mathbf{x} + b_{1R}) + \cos(\boldsymbol{\omega}_{2R}^\top \mathbf{x} + b_{2R})]^\top,$$
(8)

where $\{b_{1r}, b_{2r}\}_{r=1}^R$ are uniformly sampled from $[0, 2\pi]$. The derivation of Eq. (8) is provided in Appendix B.

The spectral feature mapping in Eq. (8) can be viewed as a more complex single-layer neural network: it employs two sets of weights ($\boldsymbol{\omega}$) and biases ($b$) to linearly transform the input, followed by the application of cosine activation functions, and then adds the results to obtain a single output dimension. By stacking multiple layers on top of each other, we naturally construct a deep nonstationary kernel:

$$\Psi^{(R)}(\mathbf{x}) = \varphi_L^{(R)} \circ \varphi_{L-1}^{(R)} \circ \cdots \circ \varphi_1^{(R)}(\mathbf{x}), \quad k(\mathbf{x}_1, \mathbf{x}_2) \approx \Psi^{(R)}(\mathbf{x}_1)^\top \Psi^{(R)}(\mathbf{x}_2),$$
(9)

where $\Psi^{(R)}(\mathbf{x})$ is the spectral feature mapping with $L$ layers and $\varphi_l^{(R)}$ denotes the $l$-th layer[§]. This actually recovers the deep spectral kernel proposed by [36]. If the GP's kernel in the permanental process (Eq. (2)) adopts the above deep sparse spectral nonstationary kernel, then we refer to this model as DNSSPP.

DNSSPP exhibits enhanced expressiveness compared to NSSPP due to its deep architecture. In subsequent sections, we consistently use $\Psi^{(R)}$ as the spectral feature mapping. When it has a single layer, the model corresponds to NSSPP; when multiple layers, the model corresponds to DNSSPP.

## 5 Inference

For the posterior inference of (D)NSSPP, this work employs a fast Laplace approximation exploiting the sparse spectral representation of nonstationary kernels. Other inference methods, such as variational inference, are also feasible. Specifically, we extend the method proposed in [29] from stationary kernels to nonstationary kernels.

### 5.1 Joint Distribution

When we approximate a kernel using Eq. (6) or Eq. (9), the GP prior can also be approximated as:

$$f(\mathbf{x}) \approx \boldsymbol{\beta}^\top \Psi^{(R)}(\mathbf{x}), \quad \boldsymbol{\beta} \sim \mathcal{N}(\mathbf{0}, \mathbf{I}).$$
(10)

Additionally, we set the intensity function as $\lambda(\mathbf{x}) = (f(\mathbf{x}) + \alpha)^2$. Following [14], we introduce an offset $\alpha$ to mitigate the nodal line problem caused by the non-injectiveness of $\lambda = f^2$. This issue results in the intensity between positive and negative values of $f$ being forced to zero, despite the underlying intensity being positive.

Combing the likelihood in Eq. (1) and the equivalent GP prior in Eq. (10), we obtain the joint density:

$$\begin{aligned}
\log p(\{\mathbf{x}_i\}_{i=1}^N, \boldsymbol{\beta}|\Theta) &= \log p(\{\mathbf{x}_i\}_{i=1}^N |\boldsymbol{\beta}, \Theta) + \log p(\boldsymbol{\beta}) \\
&= \sum_{i=1}^N \log((\boldsymbol{\beta}^\top \Psi^{(R)}(\mathbf{x}_i) + \alpha)^2) - \int_\mathcal{X} \lambda(\mathbf{x}) d\mathbf{x} - \frac{1}{2}\boldsymbol{\beta}^\top \boldsymbol{\beta} + C,
\end{aligned}$$
(11)

where $C$ is a constant and $\Theta$ denotes the model hyperparameters. The intensity integral term can be further expressed as:

$$\int_\mathcal{X} \lambda(\mathbf{x}) d\mathbf{x} = \boldsymbol{\beta}^\top \mathbf{M} \boldsymbol{\beta} + 2\alpha \boldsymbol{\beta}^\top \mathbf{m} + \alpha^2 |\mathcal{X}|,$$
(12)

---

[§]For simplicity, here we assume all layers have the same width $R$, but it can also vary.

where $|\mathcal{X}|$ is the window size, $\mathbf{M}$ is a $R \times R$ matrix, and $\mathbf{m}$ is a $R$-sized vector with each entry:

$$\mathbf{M}_{i,j} = \int_{\mathcal{X}} \Psi_i^{(R)}(\mathbf{x})\Psi_j^{(R)}(\mathbf{x})d\mathbf{x}, \quad \mathbf{m}_i = \int_{\mathcal{X}} \Psi_i^{(R)}(\mathbf{x})d\mathbf{x}, \quad i,j \in 1,\ldots,R. \tag{13}$$

It is worth noting that the intensity integral is potentially analytically solvable. For a single-layer spectral feature mapping (NSSPP), analytical solutions for $\mathbf{M}$ and $\mathbf{m}$ exist (see Appendix C). However, for multiple-layer mapping (DNSSPP), analytical solutions are not possible. In the following, we analytically solve $\mathbf{M}$ and $\mathbf{m}$ for NSSPP, while using numerical integration for DNSSPP.

## 5.2  Laplace Approximation

We use the Laplace's method to approximate the posterior $p(\boldsymbol{\beta}|\{\mathbf{x}_i\}_{i=1}^{N}, \Theta)$. The Laplace's method approximates the posterior with a Gaussian distribution by performing a second-order Taylor expansion around the maximum of the log posterior. Following the common practice:

$$\log p(\boldsymbol{\beta}|\{\mathbf{x}_i\}, \Theta) \approx \log \mathcal{N}(\boldsymbol{\beta}|\hat{\boldsymbol{\beta}}, \mathbf{Q}) = -\frac{1}{2}(\boldsymbol{\beta} - \hat{\boldsymbol{\beta}})^{\top}\mathbf{Q}^{-1}(\boldsymbol{\beta} - \hat{\boldsymbol{\beta}}) - \frac{1}{2}\log|\mathbf{Q}| - \frac{D}{2}\log 2\pi, \tag{14}$$

where $\hat{\boldsymbol{\beta}}$ is the mode of the true posterior and $\mathbf{Q}$ is the negative inverse Hessian of the true posterior evaluated at the mode:

$$\nabla_{\boldsymbol{\beta}} \log p(\{\mathbf{x}_i\}_{i=1}^{N}, \boldsymbol{\beta}|\Theta)|_{\boldsymbol{\beta}=\hat{\boldsymbol{\beta}}} = 0, \quad \mathbf{Q}^{-1} = -\nabla_{\boldsymbol{\beta}}^2 \log p(\{\mathbf{x}_i\}_{i=1}^{N}, \boldsymbol{\beta}|\Theta)|_{\boldsymbol{\beta}=\hat{\boldsymbol{\beta}}}. \tag{15}$$

The mode $\hat{\boldsymbol{\beta}}$ is typically not directly solvable based on Eq. (15), so we employ optimization methods to find it. Then, the precision matrix $\mathbf{Q}^{-1}$ can be computed analytically (see Appendix D).

## 5.3  Hyperparameter and Complexity

The model hyperparameter $\Theta$ consists of the kernel parameters $\sigma, \boldsymbol{\omega}, \mathbf{b}$ (if there are multiple layers, these hyperparameters include parameters from all layers) and the bias of the intensity function $\alpha$. We can learn these hyperparameters from the data by maximizing the marginal likelihood:

$$\log p(\{\mathbf{x}_i\}_{i=1}^{N}|\Theta)$$
$$= \log p(\{\mathbf{x}_i\}_{i=1}^{N}, \boldsymbol{\beta}|\Theta) - \log p(\boldsymbol{\beta}|\{\mathbf{x}_i\}_{i=1}^{N}, \Theta) \approx \log p(\{\mathbf{x}_i\}_{i=1}^{N}, \boldsymbol{\beta}|\Theta) - \log \mathcal{N}(\boldsymbol{\beta}|\hat{\boldsymbol{\beta}}, \mathbf{Q}). \tag{16}$$

Substituting Eqs. (11) and (14) into Eq. (16), we can obtain a marginal likelihood approximation. Optimizing this yields the optimal hyperparameter $\Theta$.

Similar to common low-rank approximation methods for kernels, the computational complexity of the proposed inference method is reduced from $O(N^3)$ to $O(NR^2)$, where $R \ll N$, i.e., linearly with $N$. A complete algorithm pseudocode is provided in Appendix E.

## 5.4  Predictive Distribution

After obtaining the posterior $\boldsymbol{\beta}|\{\mathbf{x}_i\}_{i=1}^{N}, \Theta \sim \mathcal{N}(\boldsymbol{\beta}|\hat{\boldsymbol{\beta}}, \mathbf{Q})$, we can compute the posterior of the intensity function. For any $\mathbf{x}^* \in \mathcal{X}$, the predictive distribution of $f(\mathbf{x}^*) + \alpha$ is

$$f(\mathbf{x}^*) + \alpha|\{\mathbf{x}_i\}_{i=1}^{N}, \Theta \sim \mathcal{N}(\mu(\mathbf{x}^*), \sigma^2(\mathbf{x}^*)),$$

where

$$\mu(\mathbf{x}^*) = \hat{\boldsymbol{\beta}}^{\top}\Psi^{(R)}(\mathbf{x}^*) + \alpha, \quad \sigma^2(\mathbf{x}^*) = \Psi^{(R)}(\mathbf{x}^*)^{\top}\mathbf{Q}\Psi^{(R)}(\mathbf{x}^*).$$

Then, the intensity $\lambda(\mathbf{x}^*) = \left(f(\mathbf{x}^*) + \alpha\right)^2$ can be proven to have the following mean and variance:

$$\mathbb{E}[\lambda(\mathbf{x}^*)] = \mu^2(\mathbf{x}^*) + \sigma^2(\mathbf{x}^*), \quad \text{Var}[\lambda(\mathbf{x}^*)] = 2\sigma^4(\mathbf{x}^*) + 4\mu^2(\mathbf{x}^*)\sigma^2(\mathbf{x}^*). \tag{17}$$

# 6  Experiments

In this section, we mainly analyze the superiority in performance of (D)NSSPP over baseline models on both synthetic and real-world datasets, as well as the impact of various hyperparameters. We perform all experiments using the server with two GPUs (NVIDIA TITAN V with 12GB memory), two CPUs (each with 8 cores, Intel(R) Xeon(R) CPU E5-2620 v4 @ 2.10GHz), and 251GB memory.

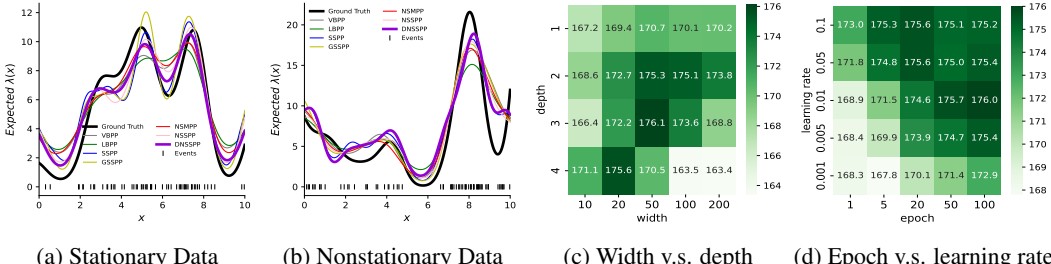

| (a) Stationary Data | (b) Nonstationary Data | (c) Width v.s. depth | (d) Epoch v.s. learning rate |

Figure 1: (a) The fitting results of the intensity functions for all models on the stationary synthetic data; (b) those on the nonstationary synthetic data. The impact of (c) network width and depth, (d) the number of epochs and learning rate on the $\mathcal{L}_{\text{test}}$ of DNSSPP on the nonstationary data.

## 6.1 Baselines

We employ several previous works as baselines for comparison. These baselines include **VBPP** which utilizes the inducing points method [18], **LBPP** employing the Nyström approximation [33], as well as **SSPP** and **GSSPP** utilizing the Fourier representation [29]. We use GSSPP based on Gaussian kernel, Matérn kernel with parameter 1/2 and 5/2 respectively, and we denote them as GSSPP, GSSPP-M12 and GSSPP-M52. Besides, we offer a baseline that employs stacked mappings of stationary kernels which we name it Deep Sparse Spectral Permanental Process (**DSSPP**). All of these methods utilize stationary kernels. As far as we know, there is currently no work employing nonstationary kernels in permanental processes. To compare with nonstationary baselines, we implement a nonstationary permanental process using the nonstationary spectral mixture kernel [26], referred to as **NSMPP**.

## 6.2 Metrics

We employ two metrics to evaluate the performance of various baselines. One is the expected log-likelihood on the test data, denoted as $\mathcal{L}_{\text{test}}$, with details provided in Appendix F. Additionally, when the ground-truth intensity is available, e.g., for the synthetic data, the root mean square error (**RMSE**) between the expected posterior intensity and the ground truth serves as another measure.

## 6.3 Synthetic Data

**Datasets**   To better compare the performance of (D)NSSPP with baseline models on point process data in both stationary and nonstationary scenarios, we simulated a stationary and a nonstationary permanental process on the interval $[0, 10]$, respectively. Specifically, we assume two kernels:

$$k_1(\mathbf{x}_1, \mathbf{x}_2) = \exp(-\frac{1}{2}\|\mathbf{x}_1 - \mathbf{x}_2\|^2), \quad k_2(\mathbf{x}_1, \mathbf{x}_2) = (\frac{\mathbf{x}_1^\top \mathbf{x}_2}{100} + 1)^3 \exp(-\frac{1}{2}\|\mathbf{x}_1 - \mathbf{x}_2\|^2),$$

where $k_1$ is a stationary Gaussian kernel and $k_2$ is a nonstationary kernel obtained by multiplying a Gaussian kernel with a polynomial kernel. We use these two kernels to construct a stationary GP and a nonstationary GP, respectively, and randomly sample two corresponding latent functions. Based on $\lambda(\mathbf{x}) = (f(\mathbf{x}) + 2)^2$, we construct two corresponding intensity functions. Finally, we use the thinning algorithm [22] to simulate two sets of synthetic data. For each intensity, we simulate ten datasets and use each dataset alternately as the training set and the remaining ones as the test sets.

**Setup**   For NSSPP, the number of frequencies (network width) is set to 50. For DNSSPP, we experimented with three different configurations: DNSSPP-[50,30], DNSSPP-[100,50], and DNSSPP-[30,50,30]. Each number represents the width of the corresponding network layer. Therefore, DNSSPP-[50,30] implies a model with two layers, where the first layer has a width of 50 and the second layer has a width of 30, and so on. For baseline models, we need to set the number of rank for kernel low-rank approximation. We set the number of inducing points to 50 for VBPP, the number of eigenvalues to 50 for LBPP, and the number of frequencies to 50 for SSPP, GSSPP, and NSMPP. For DSSPP, to facilitate comparison with DNSSPP, we adopt the same layer configurations.

**Results**   The fitting results of intensity functions for all models are depicted in Fig. 1. To quantitatively compare the fitting performance of different methods in both stationary and nonstationary

Table 1: The performance of $\mathcal{L}_{\text{test}}$, RMSE and runtime for DNSSPP, DSSPP, NSSPP and other baselines on two synthetic datasets. For $\mathcal{L}_{\text{test}}$, the higher the better; for RMSE and runtime, the lower the better. The upper half corresponds to nonstationary models, while the lower half to stationary models. On the stationary dataset, DSSPP performs comparably to DNSSPP. On the nonstationary dataset, DSSPP does not outperform DNSSPP due to the severe nonstationarity in the data.

| | Stationary Synthetic Data | | | Nonstationary Synthetic Data | | |
|---|---|---|---|---|---|---|
| | $\mathcal{L}_{\text{test}}$ | RMSE | Runtime(s) | $\mathcal{L}_{\text{test}}$ | RMSE | Runtime(s) |
| DNSSPP-[100,50] | 156.43($\pm$ 37.52) | 0.061($\pm$ 0.020) | 9.27 | **175.70($\pm$ 26.89)** | **0.076($\pm$ 0.015)** | 9.40 |
| DNSSPP-[50,30] | **156.55($\pm$ 37.44)** | 0.061($\pm$ 0.017) | 9.00 | 173.68($\pm$ 26.70) | 0.094($\pm$ 0.012) | 9.11 |
| DNSSPP-[30,50,30] | 156.55($\pm$ 37.66) | 0.068($\pm$ 0.017) | 10.32 | 174.47($\pm$ 26.57) | 0.086($\pm$ 0.013) | 10.07 |
| NSSPP | 153.56($\pm$ 36.82) | 0.070($\pm$ 0.009) | 8.39 | 172.87($\pm$ 26.86) | 0.10($\pm$ 0.012) | 8.96 |
| NSMPP | 153.05($\pm$ 36.96) | 0.065($\pm$ 0.018) | 3.19 | 172.17($\pm$ 26.42) | 0.079($\pm$ 0.019) | 4.63 |
| DSSPP-[100,50] | 155.62($\pm$ 37.31) | **0.060($\pm$ 0.015)** | 7.64 | 173.91($\pm$ 10.27) | 0.090($\pm$ 0.013) | 8.27 |
| DSSPP-[50,30] | 155.51($\pm$ 36.98) | 0.065($\pm$ 0.012) | 7.56 | 172.54($\pm$ 26.74) | 0.103($\pm$ 0.015) | 7.54 |
| DSSPP-[30,50,30] | 153.75($\pm$ 37.34) | 0.073($\pm$ 0.016) | 8.95 | 174.35($\pm$ 26.70) | 0.086($\pm$ 0.013) | 9.74 |
| SSPP | 153.91($\pm$ 36.70) | 0.071($\pm$ 0.016) | 1.95 | 165.94($\pm$ 30.00) | 0.140($\pm$ 0.009) | 2.48 |
| GSSPP | 154.12($\pm$ 37.09) | 0.073($\pm$ 0.014) | 7.72 | 170.13($\pm$ 26.43) | 0.101($\pm$ 0.022) | 14.19 |
| GSSPP-M12 | 150.65($\pm$ 36.84) | 0.071($\pm$ 0.018) | 9.30 | 168.49($\pm$ 26.67) | 0.083($\pm$ 0.018) | 15.33 |
| GSSPP-M52 | 154.46($\pm$ 37.24) | 0.075($\pm$ 0.022) | 9.63 | 169.33($\pm$ 26.61) | 0.107($\pm$ 0.026) | 14.49 |
| LBPP | 150.80($\pm$ 36.13) | 0.082($\pm$ 0.008) | 0.31 | 168.97($\pm$ 26.70) | 0.126($\pm$ 0.006) | 0.37 |
| VBPP | 155.29($\pm$ 36.52) | 0.072($\pm$ 0.021) | 1.68 | 172.95($\pm$ 30.00) | 0.087($\pm$ 0.016) | 1.83 |

LBPP for Redwoods    DNSSPP for Redwoods    LBPP for Taxi    DNSSPP for Taxi

Figure 2: The fitting results of the intensity functions from LBPP and DNSSPP on the Redwoods and Taxi datasets. Additional results for various baselines on three datasets are provided in Appendix G.

scenarios, we employ $\mathcal{L}_{\text{test}}$ and RMSE as metrics. The comparison results are presented in Table 1. DNSSPP with three different configurations exhibits significant advantages on both stationary and nonstationary datasets. NSSPP and NSMPP show better performance in the nonstationary scenario compared to most stationary baselines. This is reasonable, as stationary baselines face challenges in accurately modeling nonstationary data. Because stationary kernels are a subset of nonstationary kernels, nonstationary models typically perform well on stationary data as well. Compared to relatively shallow models, DNSSPP achieves better performance but also incurs a longer running time.

### 6.4 Real Data

**Datasets**    In this section, we use three sets of common real-world datasets to evaluate the performance of our method and the baselines. **Coal Mining Disaster** [13]: This is a 1-dimensional dataset containing 191 incidents that occurred between March 15, 1875, and March 22, 1962. Each event represents a mining accident that killed more than 10 people. **Redwoods** [27]: This is a 2-dimensional dataset that describes the distribution of redwoods in an area, consisting of 195 events. **Porto Taxi** [21]: This is a large 2-dimensional dataset containing the tracks of 7,000 taxis in Porto during 2013/2014. We focus on the area with coordinates between $(41.147, -8.58)$ and $(41.18, -8.65)$, and extract 3,000 pick-up points as our dataset. For each dataset, we randomly partition the data into training set and test set of approximately equal size.

**Setup**    For real datasets, because we do not know the ground-truth intensity, we use $\mathcal{L}_{\text{test}}$ as the evaluation metric. Since NSSPP generally performs worse than DNSSPP, we report only the results of DNSSPP in this section. For DNSSPP, we consistently use the same three configurations as in the synthetic data. For baseline models, we choose different numbers of rank for kernel low-rank approximation for 1-dimensional and 2-dimensional datasets. Specifically, for the 1-dimensional dataset (Coal), we set the number of rank, i.e., the number of inducing points for VBPP, the number

Table 2: The performance of $\mathcal{L}_{\text{test}}$ and runtime for DNSSPP and other baselines on three real datasets. The upper half corresponds to nonstationary models, while the lower half to stationary models.

| | Coal | | Redwoods | | Taxi | |
|---|---|---|---|---|---|---|
| | $\mathcal{L}_{\text{test}}$ | Runtime(s) | $\mathcal{L}_{\text{test}}$ | Runtime(s) | $\mathcal{L}_{\text{test}}$ | Runtime(s) |
| DNSSPP-[100,50] | 225.73($\pm$ 2.96) | 10.54 | 79.56($\pm$ 0.014) | 11.09 | 7171($\pm$ 73) | 73.3 |
| DNSSPP-[50,30] | **225.85($\pm$ 2.61)** | 11.38 | 79.50($\pm$ 0.020) | 10.45 | 6682($\pm$ 37) | 70.1 |
| DNSSPP-[30,50,30] | 225.79($\pm$ 4.49) | 11.76 | 79.55($\pm$ 0.025) | 12.72 | **7246($\pm$ 37)** | 81.0 |
| NSMPP | 223.28($\pm$ 3.60) | 2.88 | 77.64($\pm$ 6.21) | 4.43 | 6492($\pm$ 62) | 94.6 |
| SSPP | 221.42($\pm$ 1.87) | 1.72 | 78.57($\pm$ 2.83) | 2.45 | 6245($\pm$ 45) | 58.4 |
| GSSPP | 221.08($\pm$ 6.32) | 5.05 | 76.97($\pm$ 5.80) | 5.94 | 6445($\pm$ 97) | 110.64 |
| GSSPP-M12 | 223.11($\pm$ 5.02) | 5.61 | 72.96($\pm$ 11.89) | 5.90 | 6599($\pm$ 76) | 104.75 |
| GSSPP-M52 | 221.89($\pm$ 3.06) | 5.17 | 77.98($\pm$ 6.05) | 5.23 | 6526($\pm$ 113) | 112.54 |
| LBPP | 218.30($\pm$ 4.12) | 0.33 | **80.40($\pm$ 0.72)** | 0.34 | 6096($\pm$ 25) | 3.16 |
| VBPP | 219.15($\pm$ 4.54) | 1.69 | 77.06($\pm$ 0.88) | 2.14 | 6156($\pm$ 34) | 9.10 |

of eigenvalues for LBPP, and the number of frequencies for SSPP, GSSPP, and NSMPP, to 10. For the 2-dimensional datasets (Redwoods and Taxi), we set the number of rank to 50. We discuss the reasonableness of the above setup in Appendix H.

**Results** Fig. 2 shows the fitting results of the intensity functions from LBPP and DNSSPP on the Redwoods and Taxi datasets. The quantitative comparison results are shown in Table 2. Since real-world data is more or less nonstationary to varying degrees, nonstationary models generally perform better on real-world data. DNSSPP secures both first and second places on the Coal and Taxi datasets and achieves second place on the Redwoods dataset. For simpler datasets, such as Coal and Redwoods, which have relatively simple data patterns, the performance of DNSSPP is comparable to that of the simpler baselines. However, for the more complex Taxi dataset, DNSSPP demonstrates significant advantages over the simpler baselines. This indicates that DNSSPP has a clear advantage in handling data with more complex patterns, such as large-scale or high-dimensional datasets.

## 6.5 Ablation Studies

In this section, we analyze the impact of four configuration hyperparameters on the performance of DNSSPP: the network width and depth, as well as the number of epochs and learning rate during the update of hyperparameter $\Theta$.

### 6.5.1 Width and Depth of Network

We investigate DNSSPP's performance with varying network sizes by adjusting width and depth on nonstationary synthetic data. In our experiments, we maintain consistent width across layers. Results are illustrated in Fig. 1c. As the number of layers and the width of the network increase, DNSSPP's performance initially improves and then declines, reflecting the phenomenon of overfitting as the network size increases. Regarding the number of layers, DNSSPP generally exhibits underfitting when there is only a single layer, while overfitting tends to occur with four layers. In terms of network width, due to the relatively small size of the synthetic data, overfitting becomes apparent when the width is large. However, for larger datasets, a more complex network structure should be chosen.

### 6.5.2 Epoch and Learning Rate of $\Theta$

During the inference process, we need to perform numerical optimization on the hyperparameter $\Theta$, implying the need to determine the epoch and learning rate for optimization. We conduct experiments on the nonstationary synthetic data to investigate the coordination of epoch and learning rate. Results are illustrated in Fig. 1d. When the learning rate is too low or the number of epochs is insufficient, the update of $\Theta$ is sluggish, leading to inferior performance. However, when the learning rate is too high or the number of epochs is excessive, model performance generally declines as well. This is because our model training is a bi-level optimization: the inner level is the Laplace approximation, and the outer level is to update the hyperparameter $\Theta$. An excessively high learning rate or too many epochs can cause $\Theta$ to converge too quickly, making the overall model training more prone to getting stuck in local optima, thus resulting in suboptimal performance.

# 7 Limitations

NSSPP not only removes restrictions on the kernel's functional form but also makes it applicable to nonstationary kernels. DNSSPP further enhances its expressive power through a deep architecture. However, the deep architecture of DNSSPP prevents analytical computation of the intensity integral, necessitating numerical integration. This reliance on numerical integration imposes a limitation on the model's computational speed.

# 8 Conclusions

In this study, we introduced NSSPP and its deep kernel variant, DNSSPP, to address limitations in modeling the permanental process. This approach overcomes the stationary assumption, allowing for flexible learning of nonstationary kernels directly from data without restricting their form. By leveraging the sparse spectral representation of nonstationary kernels, we achieved a low-rank approximation, effectively reducing computational complexity. Our experiments on synthetic and real-world datasets demonstrated that (D)NSSPP achieved competitive performance on stationary data while excelling on nonstationary datasets. This study underscores the importance of considering nonstationarity in the permanental process and demonstrates the efficacy of (D)NSSPP in this context.

## Acknowledgments and Disclosure of Funding

This work was supported by NSFC Projects (Nos. 62106121, 62406119), the MOE Project of Key Research Institute of Humanities and Social Sciences (22JJD110001), the fundamental research funds for the central universities, and the research funds of Renmin University of China (24XNKJ13), the Natural Science Foundation of Hubei Province (2024AFB074), and the Guangdong Provincial Key Laboratory of Mathematical Foundations for Artificial Intelligence (2023B1212010001).

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

## A  Proof of Equation (4)

The derivation of Eq. (4) is provided below:

$$
\begin{aligned}
k(\mathbf{x}_1 - \mathbf{x}_2) &\approx \frac{\sigma^2}{R} \sum_{i=1}^{R} \phi_i(\mathbf{x}_1)\phi_i(\mathbf{x}_2)^* \\
&= \frac{\sigma^2}{R} \sum_{i=1}^{R} \exp(i\boldsymbol{\omega}_i^\top \mathbf{x}_1)\exp(-i\boldsymbol{\omega}_i^\top \mathbf{x}_2) \\
&= \frac{\sigma^2}{R} \sum_{i=1}^{R} (\cos(\boldsymbol{\omega}_i^\top \mathbf{x}_1) + i\sin(\boldsymbol{\omega}_i^\top \mathbf{x}_1))(\cos(\boldsymbol{\omega}_i^\top \mathbf{x}_2) - i\sin(\boldsymbol{\omega}_i^\top \mathbf{x}_2)) \\
&= \frac{\sigma^2}{R} \sum_{i=1}^{R} (\cos(\boldsymbol{\omega}_i^\top \mathbf{x}_1)\cos(\boldsymbol{\omega}_i^\top \mathbf{x}_2) + \sin(\boldsymbol{\omega}_i^\top \mathbf{x}_1)\sin(\boldsymbol{\omega}_i^\top \mathbf{x}_2)) \\
&= \Phi^{(R)}(\mathbf{x}_1)^\top \Phi^{(R)}(\mathbf{x}_2).
\end{aligned}
\tag{18}
$$

In the derivation, due to the symmetry of the spectrum distribution, it can be assumed that $2R$ points are symmetrically sampled, which allows the elimination of the imaginary part in the equation without altering the expectation.

## B  Proof of Equation (8)

The derivation of Eq. (8) is provided below:

$$
\begin{aligned}
k(\mathbf{x}_1, \mathbf{x}_2) &\approx \frac{\sigma^2}{4R} \sum_{r=1}^{R} \left[ \exp\left(i(\boldsymbol{\omega}_{1r}^\top \mathbf{x}_1 - \boldsymbol{\omega}_{2r}^\top \mathbf{x}_2)\right) + \ldots + \exp\left(i(\boldsymbol{\omega}_{2r}^\top \mathbf{x}_1 - \boldsymbol{\omega}_{2r}^\top \mathbf{x}_2)\right) \right] \\
&= \frac{\sigma^2}{4R} \sum_{r=1}^{R} \left[ \cos(\boldsymbol{\omega}_{1r}^\top \mathbf{x}_1 - \boldsymbol{\omega}_{2r}^\top \mathbf{x}_2) + \ldots + \cos(\boldsymbol{\omega}_{2r}^\top \mathbf{x}_1 - \boldsymbol{\omega}_{2r}^\top \mathbf{x}_2) \right] \\
&= \frac{\sigma^2}{4R} \sum_{r=1}^{R} \sum_{i,j=1,2} \left[ \cos(\boldsymbol{\omega}_{ir}^\top \mathbf{x}_1 - \boldsymbol{\omega}_{jr}^\top \mathbf{x}_2) \right],
\end{aligned}
\tag{19}
$$

where the second line is because the kernel is real-valued, so we can safely eliminate the imaginary part. Additionally, considering the law of total expectation, we have

$$
\mathbb{E}_{\boldsymbol{\omega}_i, \boldsymbol{\omega}_j}[\cos(\boldsymbol{\omega}_i^\top \mathbf{x}_1 + \boldsymbol{\omega}_j^\top \mathbf{x}_2 + 2b)] = \mathbb{E}_{\boldsymbol{\omega}_i, \boldsymbol{\omega}_j}[\mathbb{E}_b[\cos(\boldsymbol{\omega}_i^\top \mathbf{x}_1 + \boldsymbol{\omega}_j^\top \mathbf{x}_2 + 2b)|\boldsymbol{\omega}_i, \boldsymbol{\omega}_j]],
\tag{20}
$$

where $b \sim \mathrm{Uniform}(0, 2\pi)$. And consider the periodicity of cosine function, we have

$$
\mathbb{E}_b[\cos(\boldsymbol{\omega}_i^\top \mathbf{x}_1 + \boldsymbol{\omega}_j^\top \mathbf{x}_2 + 2b)|\boldsymbol{\omega}_i, \boldsymbol{\omega}_j] = 0,
\tag{21}
$$

so Eq. (20) is finally equal to 0. Therefore, we can further obtain another equivalent form of the spectral feature mapping, through the following procedure:

$$
\begin{aligned}
k(\mathbf{x}_1, \mathbf{x}_2) &\approx \frac{\sigma^2}{4R} \sum_{r=1}^{R} \sum_{(i,j)=\{1,2\}^2} \left[ \cos(\boldsymbol{\omega}_{ir}^\top \mathbf{x}_1 - \boldsymbol{\omega}_{jr}^\top \mathbf{x}_2) + \cos(\boldsymbol{\omega}_{ir}^\top \mathbf{x}_1 + \boldsymbol{\omega}_{jr}^\top \mathbf{x}_2 + b_{ir} + b_{jr}) \right] \\
&= \frac{\sigma^2}{4R} \sum_{r=1}^{R} \sum_{(i,j)=\{1,2\}^2} 2\cos(\boldsymbol{\omega}_{ir}^\top \mathbf{x}_1 + b_{ir})\cos(\boldsymbol{\omega}_{jr}^\top \mathbf{x}_1 + b_{jr}) \\
&= \boldsymbol{\varphi}^{(R)}(\mathbf{x}_1)^\top \boldsymbol{\varphi}^{(R)}(\mathbf{x}_2),
\end{aligned}
\tag{22}
$$

where $\{b_{1r}, b_{2r}\}_{r=1}^{R}$ are uniformly sampled from $[0, 2\pi]$.

## C  Analytical Computation of Intensity Integral

The intensity function of the permanental process has the following expression $\lambda(\mathbf{x}) = (\boldsymbol{\beta}^\top \Psi^{(R)}(\mathbf{x}) + \alpha)^2$. For ease of derivation, we adopt the form of Eq. (7) for the spectral feature mapping $\Psi^{(R)}(\mathbf{x})$ in subsequent steps, i.e., a $2R$-sized vector. When $\Psi^{(R)}(\mathbf{x})$ takes the form of Eq. (8), i.e., an $R$-sized vector, the calculation result of the intensity integral remains the same. The intensity integral on $\mathcal{X}$ is:

$$
\begin{aligned}
\int_{\mathcal{X}} \lambda(\mathbf{x}) d\mathbf{x} &= \int_{\mathcal{X}} (\boldsymbol{\beta}^\top \Psi^{(R)}(\mathbf{x}) + \alpha)^2 d\mathbf{x} \\
&= \boldsymbol{\beta}^\top \int_{\mathcal{X}} \Psi^{(R)}(\mathbf{x}) \Psi^{(R)}(\mathbf{x})^\top d\mathbf{x} \boldsymbol{\beta} + 2\alpha \boldsymbol{\beta}^\top \int_{\mathcal{X}} \Psi^{(R)}(\mathbf{x}) d\mathbf{x} + \int_{\mathcal{X}} \alpha^2 d\mathbf{x} \\
&= \boldsymbol{\beta}^\top \mathbf{M} \boldsymbol{\beta} + 2\alpha \boldsymbol{\beta}^\top \mathbf{m} + \alpha^2 |\mathcal{X}|,
\end{aligned}
$$

where

$$
\mathbf{M}_{i,j} = \int_{\mathcal{X}} \Psi_i^{(R)}(\mathbf{x}) \Psi_j^{(R)}(\mathbf{x}) d\mathbf{x}, \quad \mathbf{m}_i = \int_{\mathcal{X}} \Psi_i^{(R)}(\mathbf{x}) d\mathbf{x}, \quad i, j \in 1, \ldots, 2R. \tag{23}
$$

It is worth noting that $\mathbf{M}$ and $\mathbf{m}$ are analytically solvable. The analytical solution for the stationary case is provided in Appendix C.2 in [29], and we provide the analytical expression for the nonstationary case here.

For $\mathbf{M}$, the product of two spectral feature mappings is:

$$
\Psi_i^{(R)}(\mathbf{x}) \Psi_j^{(R)}(\mathbf{x}) = \frac{\sigma^2}{4R}
\begin{cases}
\sum\limits_{(a,b)=\{1,2\}^2} \cos(\boldsymbol{\omega}_{ai}^\top \mathbf{x}) \cos(\boldsymbol{\omega}_{bj}^\top \mathbf{x}), & \text{if } i, j = 1, \ldots, R, \\
\sum\limits_{(a,b)=\{1,2\}^2} \sin(\boldsymbol{\omega}_{ai}^\top \mathbf{x}) \sin(\boldsymbol{\omega}_{bj}^\top \mathbf{x}), & \text{if } i, j = R+1, \ldots, 2R, \\
\sum\limits_{(a,b)=\{1,2\}^2} \cos(\boldsymbol{\omega}_{ai}^\top \mathbf{x}) \sin(\boldsymbol{\omega}_{bj}^\top \mathbf{x}), & \text{if } i = 1, \ldots, R, \ j = R+1, \ldots, 2R, \\
\sum\limits_{(a,b)=\{1,2\}^2} \sin(\boldsymbol{\omega}_{ai}^\top \mathbf{x}) \cos(\boldsymbol{\omega}_{bj}^\top \mathbf{x}), & \text{if } i = R+1, \ldots, 2R, \ j = 1, \ldots, R.
\end{cases}
\tag{24}
$$

We can further transform the product term of trigonometric functions:

$$
\begin{aligned}
\cos(\boldsymbol{\omega}_{ai}^\top \mathbf{x}) \cos(\boldsymbol{\omega}_{bj}^\top \mathbf{x}) &= \frac{1}{2}[\cos\left((\boldsymbol{\omega}_{ai} - \boldsymbol{\omega}_{bj})^\top \mathbf{x}\right) + \cos\left((\boldsymbol{\omega}_{ai} + \boldsymbol{\omega}_{bj})^\top \mathbf{x}\right)], \\
\sin(\boldsymbol{\omega}_{ai}^\top \mathbf{x}) \sin(\boldsymbol{\omega}_{bj}^\top \mathbf{x}) &= \frac{1}{2}[\cos\left((\boldsymbol{\omega}_{ai} - \boldsymbol{\omega}_{bj})^\top \mathbf{x}\right) - \cos\left((\boldsymbol{\omega}_{ai} + \boldsymbol{\omega}_{bj})^\top \mathbf{x}\right)], \\
\cos(\boldsymbol{\omega}_{ai}^\top \mathbf{x}) \sin(\boldsymbol{\omega}_{bj}^\top \mathbf{x}) &= \frac{1}{2}[\sin\left((\boldsymbol{\omega}_{ai} - \boldsymbol{\omega}_{bj})^\top \mathbf{x}\right) + \sin\left((\boldsymbol{\omega}_{ai} + \boldsymbol{\omega}_{bj})^\top \mathbf{x}\right)].
\end{aligned}
\tag{25}
$$

Without loss of generality, we consider the integral domain $\mathcal{X}$ as $[-d, d]^D$. Then we can compute the integral of the above expression analytically. Since we only consider the $D = 1$ and $D = 2$ point process data in this paper, we specifically demonstrate the analytical integral expressions in the one-dimensional and two-dimensional cases here.

One-dimension:

$$
\int_{[-d,d]} \cos(\eta x) dx =
\begin{cases}
\frac{2}{\eta} \sin(\eta d) & \text{if } \eta \neq 0 \\
2d & \text{if } \eta = 0,
\end{cases}
\tag{26}
$$

$$
\int_{[-d,d]} \sin(\eta x) dx = 0. \tag{27}
$$

Two-dimension:

$$
\int_{[-d,d]^2} \cos(\boldsymbol{\eta}^\top \mathbf{x}) d\mathbf{x} =
\begin{cases}
\frac{2}{\eta_1 \eta_2} \left( \cos((\eta_1 - \eta_2)d) - \cos((\eta_1 + \eta_2)d) \right) & \text{if } \eta_1, \eta_2 \neq 0 \\
4d^2 & \text{if } \eta_1 = \eta_2 = 0,
\end{cases}
\tag{28}
$$

$$
\int_{[-d,d]^2} \sin(\boldsymbol{\eta}^\top \mathbf{x}) d\mathbf{x} = 0, \tag{29}
$$

where $\eta_1$ denotes the first ordinate of $\boldsymbol{\eta}$, and $\eta_2$ denotes the second ordinate of $\boldsymbol{\eta}$. It should be noted that theoretically, the integral in Eq. (28) only has two possible results as explained in the right side. The analytical solution of the integral of Eq. (25) can be obtained by replacing $\boldsymbol{\eta}$ with $\boldsymbol{\omega}_{ai} - \boldsymbol{\omega}_{bj}$ or $\boldsymbol{\omega}_{ai} + \boldsymbol{\omega}_{bj}$. Note that the case $a = b$ and $i = j$ corresponds to the second case in Eq. (26) and Eq. (28), which means the computation of the diagonal entries of $\mathbf{M}$ is different from that elsewhere.

Next, we can compute $\mathbf{m}$:

$$\mathbf{m}_i = \begin{cases} \int_{[-d,d]^D} \big( \cos(\boldsymbol{\omega}_{1i}^\top \mathbf{x}) + \cos(\boldsymbol{\omega}_{2i}^\top \mathbf{x}) \big) d\mathbf{x}, & \text{if } i = 1, \ldots, R, \\ \int_{[-d,d]^D} \big( \sin(\boldsymbol{\omega}_{1i}^\top \mathbf{x}) + \sin(\boldsymbol{\omega}_{2i}^\top \mathbf{x}) \big) d\mathbf{x}, & \text{if } i = R+1, \ldots, 2R. \end{cases} \tag{30}$$

It is easy to see that if $D = 1$, $\mathbf{m}$ can be analytically computed using Eq. (26) and Eq. (27), while if $D = 2$, $\mathbf{m}$ can be analytically computed using Eq. (28) and Eq. (29). It is worth noting that the analytical expression of the intensity integral above applies only to NSSPP (single-layer network). For DNSSPP (multi-layer network), the nested structure of trigonometric functions leads to $\mathbf{M}$ and $\mathbf{m}$ lacking analytical solutions, and we need resort to numerical integration.

## D   Derivation of Laplace Approximation

The gradient and the Hessian matrix of the posterior of $\boldsymbol{\beta}$ are:

$$\nabla_{\boldsymbol{\beta}} \log p(\{\mathbf{x}_i\}_{i=1}^N, \boldsymbol{\beta}|\Theta) = -(2\mathbf{M} + \mathbf{I})\boldsymbol{\beta} - 2\alpha\mathbf{m} + 2\sum_{i=1}^N \frac{\Psi^{(R)}(\mathbf{x}_i)}{\boldsymbol{\beta}^\top \Psi^{(R)}(\mathbf{x}_i) + \alpha}, \tag{31}$$

$$\nabla_{\boldsymbol{\beta}}^2 \log p(\{\mathbf{x}_i\}_{i=1}^N, \boldsymbol{\beta}|\Theta) = -(2\mathbf{M} + \mathbf{I}) - 2\sum_{i=1}^N \frac{\Psi^{(R)}(\mathbf{x}_i)\Psi^{(R)}(\mathbf{x}_i)^\top}{(\boldsymbol{\beta}^\top \Psi^{(R)}(\mathbf{x}_i) + \alpha)^2}. \tag{32}$$

In theory, setting Eq. (31) to $0$ can solve for the mode of the posterior, denoted as $\hat{\boldsymbol{\beta}}$. However, in practice, we cannot analytically solve it, so we use numerical optimization to obtain $\hat{\boldsymbol{\beta}}$. Then according to Eq. (32), we can obtain the precision matrix $\mathbf{Q}^{-1} = -\nabla_{\boldsymbol{\beta}}^2 \log p(\{\mathbf{x}_i\}_{i=1}^N, \boldsymbol{\beta}|\Theta)|_{\boldsymbol{\beta}=\hat{\boldsymbol{\beta}}}$.

## E   Pseudocode

The pseudocode of our inference algorithm is provided in Algorithm 1.

---
**Algorithm 1:** Inference for (D)NSSPP
---
Define the network depth $L$ and width $R$, and initialize the hyperparameter $\Theta$;
**for** *Iteration* **do**
    Calculate $\mathbf{M}$ and $\mathbf{m}$ by Eq. (13);
    Update the mode $\hat{\boldsymbol{\beta}}$ by maximizing Eq. (14);
    Update the covariance matrix $\mathbf{Q}$ by Eq. (15);
    Update the hyperparameters $\Theta$ by maximizing Eq. (16);
**end**
For any $\mathbf{x}^*$, output the predicted mean and variance of $\lambda(\mathbf{x}^*)$ by Eq. (17).

---

## F   Computation of Expected Test Log-likelihood

The computation of expected test log-likelihood is provided in Appendix E.2 in [29]. For the completeness of the article, we restate the specific derivation process here. Considering the posterior of the intensity function, we have the approximation of the expected test log-likelihood:

$$\mathbb{E}_{\boldsymbol{\beta}}[\log p(\{\mathbf{x}_i^*\}_{i=1}^{N^*} \,|\, \{\mathbf{x}_i\}_{i=1}^N)]$$

$$\approx - \mathbb{E}_{\boldsymbol{\beta}}\big[\int_{\mathcal{X}} (\boldsymbol{\beta}^\top \Psi^{(R)}(\mathbf{x}) + \alpha)^2 d\mathbf{x}\big] + \sum_{i=1}^{N^*} \mathbb{E}_{\boldsymbol{\beta}}[\log(\boldsymbol{\beta}^\top \Psi^{(R)}(\mathbf{x}_i^*) + \alpha)^2], \tag{33}$$

derived from Eq. (1).

First, we compute the first term in Eq. (33):

$$\mathbb{E}_{\boldsymbol{\beta}}[\int_{\mathcal{X}}(\boldsymbol{\beta}^\top\Psi^{(R)}(\mathbf{x})+\alpha)^2 d\mathbf{x}] = \int_{\mathcal{X}}\mathbb{E}_{\boldsymbol{\beta}}[\boldsymbol{\beta}^\top\Psi^{(R)}(\mathbf{x})+\alpha]^2 d\mathbf{x} + \int_{\mathcal{X}}\mathrm{Var}[\boldsymbol{\beta}^\top\Psi^{(R)}(\mathbf{x})]d\mathbf{x}$$

$$= \int_{\mathcal{X}}(\Psi^{(R)}(\mathbf{x})^\top\boldsymbol{\beta}\boldsymbol{\beta}^\top\Psi^{(R)}(\mathbf{x}))d\mathbf{x} + 2\alpha\int_{\mathcal{X}}(\boldsymbol{\beta}^\top\Psi^{(R)}(\mathbf{x}))d\mathbf{x} + \alpha^2|\mathcal{X}| + \int_{\mathcal{X}}(\Psi^{(R)}(\mathbf{x})^\top\mathbf{Q}\Psi^{(R)}(\mathbf{x}))d\mathbf{x}$$

$$= \boldsymbol{\beta}^\top\mathbf{M}\boldsymbol{\beta} + \mathrm{tr}(\mathbf{QM}) + 2\alpha\boldsymbol{\beta}^\top m + \alpha^2|\mathcal{X}|, \tag{34}$$

where $\mathbf{M}$ and $\mathbf{m}$ are defined as Eq. (13).

Then, we consider the second term in Eq. (33). Applying the computational method used by [18], we can obtain the following expression:

$$\sum_{i=1}^{N^*}\mathbb{E}_{\boldsymbol{\beta}}[\log(\boldsymbol{\beta}^\top\Psi^{(R)}(\mathbf{x}_i^*)+\alpha)^2] \tag{35}$$

$$= -\tilde{G}(-\frac{\boldsymbol{\beta}^\top\Psi^{(R)}(\mathbf{x})+\alpha}{2\Psi^{(R)}(\mathbf{x})^\top\mathbf{Q}\Psi^{(R)}(\mathbf{x})}) + \log(\frac{1}{2}\Psi^{(R)}(\mathbf{x})^\top\mathbf{Q}\Psi^{(R)}(\mathbf{x})) - \mathrm{Const}, \tag{36}$$

where $\mathrm{Const} \approx 0.57721566$ is the Euler-Mascheroni constant. $\tilde{G}$ is defined as

$$\tilde{G}(z) = 2z\sum_{j=0}^{\infty}\frac{j!z^j}{(2)_j(1/2)_j},$$

where $(\cdot)_j$ denotes the rising Pochhammer series:

$$(a)_0 = 1, (a)_k = a(a+1)(a+2)...(a+k+1).$$

In practical computation, we first prepare a lookup table for the $\tilde{G}$ function and then obtain numerical approximations through linear interpolation.

## G  Additional Experimental Results

In this section, we provide additional experimental results for the real data. Specifically, we present the fitting results of the intensity functions from all models on the Coal dataset in Fig. 3; the fitting results of the intensity functions from VBPP, SSPP, GSSPP, and NSMPP on the Redwoods dataset in Fig. 4; and the fitting results of the intensity functions from VBPP, SSPP, GSSPP, and NSMPP on the Taxi dataset in Fig. 5.

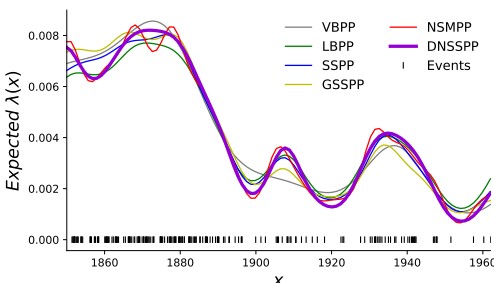

Figure 3: The fitting results of the intensity functions from all models on the Coal dataset.

## H  Some Discussion about Experimental Setup for Real Data

For the real data, we chose different numbers of ranks for the 1-dimensional dataset (10 for Coal) and the 2-dimensional datasets (50 for Redwoods and Taxi) for all baseline methods, except for DNSSPP. The reason is that the Coal dataset is very small (only 191 events) and has a simple data pattern. A

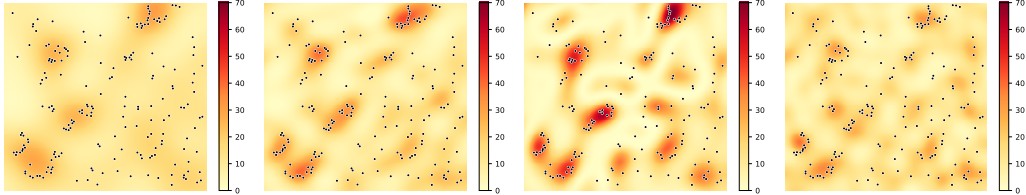

| VBPP for Redwoods | SSPP for Redwoods | GSSPP for Redwoods | NSMPP for Redwoods |

Figure 4: The fitting results of the intensity functions from VBPP, SSPP, GSSPP and NSMPP on the Redwoods dataset.

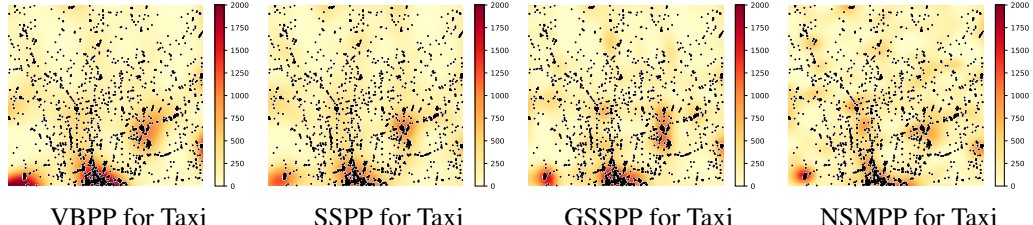

| VBPP for Taxi | SSPP for Taxi | GSSPP for Taxi | NSMPP for Taxi |

Figure 5: The fitting results of the intensity functions from VBPP, SSPP, GSSPP and NSMPP on the Taxi dataset.

higher number of ranks would not further improve performance but would significantly increase the algorithm's running time. Therefore, we set the number of ranks to 10 for Coal, but 50 for Redwoods and Taxi because the latter two datasets are larger and more complex, requiring a higher number of ranks to improve performance.

To demonstrate this, we further increased the number of ranks for all baseline models to 50 on the Coal dataset and compared the result to rank=10. The result is shown in Table 3. As can be seen from the results, when we increased the number of ranks from 10 to 50, the performance of all baseline models did not improve significantly, but the algorithm's running time increased substantially.

In summary, for the same dataset, the number of ranks for all baseline methods is kept consistent, except for DNSSPP because it is not a single-layer architecture. The choice of the number of ranks for the baseline methods was made considering the trade-off between performance and running time. Further increasing the number of ranks would not significantly improve model performance but would result in a substantial increase in computation time.

Table 3: The performance of $\mathcal{L}_{\text{test}}$ and runtime for all baselines on the Coal dataset, with the number of rank set to 10 and 50. When we increase the number of ranks from 10 to 50, the performance of all baseline models does not improve significantly, but the algorithm's running time increases substantially.

|  | Coal (rank = 10) | | Coal (rank = 50) | |
|  | $\mathcal{L}_{\text{test}}$ | Runtime(s) | $\mathcal{L}_{\text{test}}$ | Runtime(s) |
| --- | --- | --- | --- | --- |
| NSMPP | 223.28($\pm$ 3.60) | 2.88 | 222.92($\pm$ 3.63) | 4.74 |
| SSPP | 221.42($\pm$ 1.87) | 1.72 | 221.77($\pm$ 2.92) | 3.16 |
| GSSPP | 221.08($\pm$ 6.32) | 5.05 | 221.67($\pm$ 4.28) | 19.08 |
| GSSPP-M12 | 223.11($\pm$ 5.02) | 5.61 | 219.33($\pm$ 4.25) | 18.88 |
| GSSPP-M52 | 221.89($\pm$ 3.06) | 5.17 | 217.80($\pm$ 5.25) | 16.94 |
| LBPP | 218.30($\pm$ 4.12) | 0.33 | 219.88($\pm$ 2.68) | 2.68 |
| VBPP | 219.15($\pm$ 4.54) | 1.69 | 219.25($\pm$ 4.51) | 6.26 |

