# OpenReview forum: "Nonstationary Sparse Spectral Permanental Process"
_NeurIPS.cc/2024/Conference — NeurIPS 2024 poster_

### Official Review · Reviewer_vqtT · 2024-07-02

**Soundness:** 3
**Presentation:** 4
**Contribution:** 2
**Rating:** 6
**Confidence:** 3

**Summary:**

Point process are generative models for datasets of points in (typically 1D, 2D, 3D Euclidean) space, e.g. datasets of rain drops, taxi pickup locations, and a Poisson process consists of an intensity function $ \lambda: \mathcal{X} \to \mathbb{R}^+ $ that roughly shows how likely a point is to appear at $x$. The number of points $N_S$ in a region $S\subset \mathcal {X}$ is Poisson distributed $N_S\sim \text{Poi}[\lambda_S]$ with mean parameter $\lambda_S = \int_S \lambda (x) dx$.

A Poisson Process with a (squared for positive output) Gaussian Process prior over $\lambda(x)$ is a permanental process, from point clouds in 1D, 2D or 3D space, one can learn smooth underlying intensity functions. There has been much study on fitting GPs to learn intensity functions using well known GP tools such as inducing point methods and this work builds closely upon the work of [27] which uses the popular method of Random Fourier features for stationary kernels.

This paper makes two novel contributions, firstly using Random Fourier features for non-stationary kernels and  deriving the corresponding approximate marginal likelihood. Then secondly using deep kernel by nesting Fourier features that increases model capacity at the cost of requiring using numerical integrations for the approximate marginal likelihood. Experiments are performed and positive a range of results and model hyperparameters showing results both with and without improvements over baselines.

[27] [Sellier 2023](https://proceedings.mlr.press/v206/sellier23a/sellier23a.pdf)

**Strengths:**

- the idea seems a nice intuitive extension of prior work
- I felt the writing was very clear and concise and introduced all the necessary background and new ideas very clearly.
- the method and experiments appear to be well designed
- the new method DNSSPP introduces more parameters and the authors provide parameter sweeps and show results with and without improvement over baselines.

**Weaknesses:**

My only concern is the lack of substantial novel contribution, particularly when compared to the work of [27] that introduced GSSPP.

- Replacing the approximation based on Bochner's Theorem with the approximation based on Yanglon 1987 [37] significantly increases model capacity and flexibility but, unless I am mistaken, appears to be a rather small change from GSSPP, we sample over pairs of frequencies instead.
- consequently, deriving the Laplace approximate marginal Likelihood for NSSPP also appears to be not too difficult given the result for GSSPP, use trig identities to combine frequencies and I believe the results follow from the stationary case.
- using deep features understandably helps model fitting however the analytic result of NSSPP is lost.

Overall, this felt like GSSPP with different kernels, which introduced some difficulties that seemingly had fairly simple solutions.


I feel that it is an elegant although not-too-difficult extension of GSSPP and I do not feel the contribution is significant enough for publication at NeurIPS. I enjoyed the paper and I feel the contribution is certainly very respectable and well designed and demonstrates the efficacy of the method.


[27] [Sellier 2023](https://proceedings.mlr.press/v206/sellier23a/sellier23a.pdf)

[33] [Fast Bayesian Intensity Estimation for the Permanental Process, Walder and Bishop, ICML 2017](https://proceedings.mlr.press/v70/walder17a.html)

[37] [Correlation Theory of Stationary and Related Random Functions, Akiv aYaglom](https://link.springer.com/book/9781461290865)

**Questions:**

- how is the marginal likelihood for DNSSPP computed? (The paper just says "numerically" and I couldn't see details in the appendix)
- do the error bars represent a one/two standard errors or standard deviation over test set likelihoods?

**Limitations:**

The authors have described the method in full.

---

> ### Author Rebuttal · Authors · 2024-08-05
>
> > Q: My only concern is the lack of substantial novel contribution......
> I feel that it is an elegant although not-too-difficult extension of GSSPP and I do not feel the contribution is significant enough for publication at NeurIPS. I enjoyed the paper and I feel the contribution is certainly very respectable and well designed and demonstrates the efficacy of the method.
>
> A: The reviewers' main concern is that the method proposed in this paper seems "not-too-difficult" and therefore not very "novel". However, we politely disagree that a sufficiently complex method is the only significant contribution for publication at NeurIPS.
>
> Our method may appear straightforward because we have presented it in a concise manner. From a high-level perspective, our main contributions are: (1) We used a nonstationary sparse spectral kernel to replace the stationary sparse spectral kernel, thereby constructing NSSPP, (2) we then derived the corresponding Laplace approximation inference algorithm, and (3) subsequently, we built a corresponding deep variant (DNSSPP) by stacking spectral feature mappings to further enhance expressiveness.
>
> To the best of our knowledge, there has been no prior work applying (deep) nonstationary kernels in Gaussian Cox processes. This gap is what this work seeks to address. Each of these three steps above involves greater challenges when extending from the stationary case to the nonstationary case, both in theoretical derivation and algorithm implementation. Especially the third step, the deep variant (DNSSPP), has never appeared in previous Gaussian Cox processes work. This idea cleverly utilizes the relationship between spectral feature mapping and deep models, resulting in its design. This contribution has also been praised by the other two reviewers. For example, Reviewer WJmT stated: "The introduction of a deep kernel variant, DNSSPP, which stacks multiple spectral feature mappings to enhance the expressiveness of the model significantly, is another novel idea." Reviewer QimU commented: "In terms of originality and significance, this paper fills a clear gap in the existing literature on the topic, namely probing the usefulness of deep models in the permanental process setting. Whilst conceptually not a particularly complicated extension, there are obviously various nuances to moving to a deep kernel setting which are outlined clearly by the authors throughout the work."
>
> We understand that different people may have different definitions of "novelty", which may be difficult to reconcile. We greatly respect your review and hope that the above response can lead you to reconsider the cleverly designed ideas in our work. We sincerely hope you can increase the rating.
>
> > Q: How is the marginal likelihood for DNSSPP computed? (The paper just says "numerically" and I couldn't see details in the appendix).
>
> A: For DNSSPP, the nested structure of trigonometric functions leads to the intensity integral ($\mathbf{M}$ and $\mathbf{m}$) lacking an analytical solution, and we need to resort to numerical integration. Numerical integration has many choices, such as Monte Carlo integration or quadrature methods. In this work, we used Gaussian quadrature.
>
> > Q: Do the error bars represent a one/two standard errors or standard deviation over test set likelihoods?
>
> A: It is one standard deviation.

---

> ### Comment · Reviewer_vqtT · 2024-08-09
> **Thank you for the response**
>
> Thank you for the response.
>
> It seems all reviewers agree on technical aspects and only I differ by my subjective judgement on the impact for which it seems I may have been too harsh!
>
> I am have to raised my score.

---

> > ### Author Response · Authors · 2024-08-10
> > **Thanks**
> >
> > Thank you very much for your recognition and support. We will revise the paper according to your suggestions. Thank you once again for your constructive feedback and for increasing your rating.

---

### Official Review · Reviewer_QimU · 2024-07-08

**Soundness:** 3
**Presentation:** 3
**Contribution:** 3
**Rating:** 7
**Confidence:** 3

**Summary:**

Point process models are a commonly used technique for analysis of event-based data. Gaussian Cox processes are an example which use GPs to model the intensity function in a Cox process, which itself is a specific case of a Poisson process where the intensity function is a stochastic process. Generally, these types of processes are used to model rates of events occurring at given input locations. The link function within the Gaussian Cox process is a design choice; if we choose a square link function and a stationary kernel for the GP prior, this yields a stationary permanental process. The authors aim to address three key problems with these existing approaches; the cubic complexity of exact GP inference, the requirement to use kernels which allow for analytical solutions to the intensity integral, and the reliance on shallow kernels which limit the expressiveness and flexibility of the model. Introduced is a new form of permanental process model which uses a sparse spectral representation to improve computational efficiency, allows for nonstationary kernels to be used without restriction on the form of said kernel, and allows deep kernels to be formed by composing spectral feature mappings.

**Strengths:**

-	In terms originality and significance, this paper fills a clear gap in the existing literature on the topic, namely probing the usefulness of deep models in the permanental process setting. Whilst conceptually not a particularly complicated extension, there are obviously various nuances to moving to a deep kernel setting which are outlined clearly by the authors throughout the work.
-	The experimental evaluation overall is thorough, and the results are well presented in a tidy and accessible manner, with clear figures. Particularly appreciated is the inclusion of the ablation study in the main text, which is very useful information to a reader/practitioner.

**Weaknesses:**

-	This is just a reoccurring minor typo but I’m fairly certain it’s the Porto taxi dataset (as in the city of Porto, Portugal), not Proto, so just edit all mentions of this.
-	I wasn’t entirely sure of the rationale behind some of the baseline selections, for example, for some of the real-world experiments, you use as low as 10 inducing points/frequencies for the baselines, but use more than this for the DNSSPP (‘same configurations as for the synthetic data’). It would be good to have some discussion of this, and/or how the performance of the proposed methods compares to that of the baselines as we increase the number of parameters. Obviously what constitutes a “fair” comparison between models is highly subjective and depends on various factors such as computational complexity as well as parameter count, but I just wanted a little more context on this.
-	I’m aware that there has been some work by  on understanding and exploring overfitting in the context of deep kernel learning (Promises and Pitfalls of DKL, Ober et al, 2021), it would be useful to touch on some work from this area during your discussion on overfitting just to clarify to the readers that this phenomenon is not arising solely due to the size of the datasets you are modelling.

**Questions:**

-	As mentioned earlier, I think the rationale for using of a very small number of inducing points etc. for some of the baselines needs to be clarified in the text; is it the case that if you increase this number to 50+ that the baselines begin to outperform the models proposed by the authors?
-	You mention other variational methods are feasible, was there a certain specific reason that you decided upon a Laplace approximation-based approach?
-	See earlier point about discussing the impact of the deep kernel learning framework in general on overfitting.
-	See typos in weaknesses section, please fix.

**Limitations:**

The Limitations section of the paper briefly discusses the fact that the deep formulation of the DNSSPP does not allow for analytical computation of the intensity integral, which necessitates computationally inefficient numerical integration. There are also some practical considerations/limitations discussed elsewhere such as in Section 6.5.2.

---

> ### Author Rebuttal · Authors · 2024-08-05
>
> > Q: As mentioned earlier, I think the rationale for using of a very small number of inducing points etc. for some of the baselines needs to be clarified in the text; is it the case that if you increase this number to 50+ that the baselines begin to outperform the models proposed by the authors?
>
> A: Thanks for your suggestion. For the synthetic data, we used the same number of ranks (50 inducing points, eigenvalues, frequencies, etc.) for all baseline methods, except for DNSSPP because it has many layers and different layers can have different layer widths (frequencies). For the real data, we chose different numbers of ranks for the 1-dimensional dataset (10 for Coal) and the 2-dimensional datasets (50 for Redwoods and Taxi) for all baseline methods, except for DNSSPP. The reason is that the Coal dataset is very small (only 191 events) and has a simple data pattern. A higher number of ranks would not further improve performance but would significantly increase the algorithm's running time. Therefore, we set the number of ranks to 10 for Coal, but 50 for Redwoods and Taxi because the latter two datasets are larger and more complex, requiring a higher number of ranks to improve performance.
>
> To demonstrate this, we further increased the number of ranks for all baseline models to 50 on the Coal dataset and compared the result to rank=10. The result is shown in Table 2 of the rebuttal PDF. As can be seen from the results, when we increased the number of ranks from 10 to 50, the performance of all baseline models did not improve significantly, but the algorithm’s running time increased substantially.
>
> In summary, for the same dataset, the number of ranks for all baseline methods is kept consistent, except for DNSSPP because it is not a single-layer architecture. The choice of the number of ranks for the baseline methods was made considering the trade-off between performance and running time. Further increasing the number of ranks would not significantly improve model performance but would result in a substantial increase in computation time.
>
> > Q: You mention other variational methods are feasible, was there a certain specific reason that you decided upon a Laplace approximation-based approach?
>
> A: Variational inference methods, to the best of our knowledge, require certain standard types of kernels, such as the squared exponential kernel, to ensure that the intensity integral in the likelihood has an analytical solution [18]. The advantage of the Laplace approximation is that it can analytically compute the intensity integral without restricting the types of kernels. Therefore, for NSSPP, we derived a fully analytical Laplace approximation inference algorithm.
>
> When we further extended NSSPP to the deep variant DNSSPP, the introduction of the deep architecture meant that neither variational methods nor the Laplace approximation had analytical solutions. For convenience, we continued to use the Laplace approximation method employed in NSSPP.
>
> > Q: I'm aware that there has been some work by on understanding and exploring overfitting in the context of deep kernel learning (Promises and Pitfalls of DKL, Ober et al, 2021), it would be useful to touch on some work from this area during your discussion on overfitting just to clarify to the readers that this phenomenon is not arising solely due to the size of the datasets you are modelling.
>
> A: Thanks for your suggestion. We agree with your advice. In fact, in the ablation study, we also observed some overfitting phenomena in deep kernel learning. We will provide more discussion on this issue, touching on related works from this area, in the camera ready.
>
> > Q: This is just a reoccurring minor typo but I'm fairly certain it's the Porto taxi dataset (as in the city of Porto, Portugal), not Proto, so just edit all mentions of this.
>
> A: Thanks. We will fix all these typos in the camera ready.

---

> > ### Comment · Reviewer_QimU · 2024-08-08
> > **Rebuttal response**
> >
> > I appreciate the time and effort taken by the authors on the response to my review, and for addressing each of my concerns in turn. This is a well presented, thorough and novel piece of work, and I am satisfied based on the additional results provided (which should be included in the updated manuscript) that the approach presents an improvement over existing methods. As such, I increase my rating to accept.

---

> > > ### Author Response · Authors · 2024-08-09
> > > **Thanks**
> > >
> > > Thank you very much for your thoughtful review and for taking the time to carefully consider our responses. We greatly appreciate your kind words and are pleased that you find our work to be well-presented, thorough, and novel.
> > >
> > > We will certainly include the additional results in the updated manuscript, as you suggested. Your feedback is valuable in improving the quality of our work, and we are grateful for your support.
> > >
> > > Thank you once again for your constructive feedback and for increasing your rating to accept.

---

### Official Review · Reviewer_WJmT · 2024-07-15

**Soundness:** 3
**Presentation:** 3
**Contribution:** 3
**Rating:** 6
**Confidence:** 3

**Summary:**

The paper introduces an approach to modeling permanental processes by utilizing a sparse spectral representation of nonstationary kernels, termed as Nonstationary Sparse Spectral Permanental Process (NSSPP) and its deep kernel variant (DNSSPP). This method addresses the limitations of traditional permanental processes which often require specific kernel types and assume stationarity, thus restricting the model's flexibility and computational efficiency. The deep kernel variant (DNSSPP) uses hierarchically stacked spectral feature mappings, significantly enhancing the model's ability to capture complex patterns in data. The paper presents experimental results on both synthetic and real-world datasets, showing that NSSPP and DNSSPP perform competitively on stationary data and show improvements on nonstationary datasets.

**Strengths:**

- The paper introduces a novel approach by integrating sparse spectral representation with nonstationary kernels in the context of permanental processes. The introduction of a deep kernel variant, DNSSPP, which stacks multiple spectral feature mappings to enhance the expressiveness of the model significantly, is another novel idea.
- The paper is well-structured and articulates the motivations, methodology, and findings clearly.
- Experimental results show the effectiveness of the approach when compared to baselines and other recent approaches.
- The significance of this work is considerable, given its potential impact on various fields that utilize point process models.

**Weaknesses:**

- The experimental results (Table 1) indicate that performance enhancements are observed mainly with the deep kernel variant (DNSSPP) when compared to stationary kernels. It would be beneficial for the authors to include a comparison with a baseline that employs stacked mappings of stationary kernels to more clearly ascertain the source of these improvements.

**Questions:**

- Performance comparison with a baseline that employs stacked mappings of stationary kernels

**Limitations:**

Yes.

---

> ### Author Rebuttal · Authors · 2024-08-05
>
> > Q: Performance comparison with a baseline that employs stacked mappings of stationary kernels.
>
> A: Thanks for the suggestion. A baseline that employs stacked mappings of stationary kernels is an important baseline. To show the source of the performance improvement, we have re-compared with a deep stacked stationary kernel model.
>
> The results are shown in Table 1 of the rebuttal PDF. The corresponding deep stacked stationary kernel model is referred to as DSSPP, which has the same deep architecture as DNSSPP. Therefore, DSSPP also has three implementations: DSSPP-[100,50], DSSPP-[50,30], and DSSPP-[30,50,30].
>
> As can be seen from the results, the performance of the stationary kernel model significantly improves after introducing the deep architecture. On the stationary synthetic dataset, DSSPP performs comparably to DNSSPP. On the nonstationary synthetic dataset, DSSPP does not outperform DNSSPP due to the severe nonstationarity in the data. However, on any dataset, DSSPP outperforms other shallow stationary kernel models due to the introduction of the deep architecture.
> Therefore, this demonstrates that the source of performance improvement comes from the stronger expressiveness of both the deep architecture and the nonstationary kernel.

---

### Author Rebuttal · Authors · 2024-08-05

We sincerely thank all reviewers for their efforts in providing insightful comments and constructive feedback. We are encouraged that reviewers recognize that our paper proposes an interesting extension on nonstationary permanental processes [R1,R2,R3], introduces an deep kernel variant to enhance expressiveness [R1,R2,R3], includes solid numerical experiments [R1,R2,R3], and is written very clearly and concisely [R1,R2,R3]. In the following, we address reviewers' comments point by point.

---

### Decision · Program_Chairs · 2024-09-25

**Decision:**

Accept (poster)

**Comment:**

This paper proposes an approach to using sparse spectral representations of non-stationary kernels to improve the expressiveness of permanental processes, providing more flexible modeling via less constrained kernels. The majority of the reviewers initially assessed this paper as having clear motivation, strong writing, strong novelty, effective experiment designs, promising results and considerable significance. Following the author response and discussion, all of the reviewers were in favor of accepting the paper and all concerns were addressed. The scores support an accept decision. The authors are encouraged to incorporate clarifications from the discussion and reviewer suggestions into the final manuscript.